# TOXICITY IN MULTILINGUAL MACHINE TRANSLATION AT SCALE

## ABSTRACT

Machine Translation systems can produce different types of errors, some of which get characterized as *critical* or *catastrophic* due to the specific negative impact they can have on users. Automatic or human evaluation metrics do not necessarily differentiate between such critical errors and more innocuous ones. In this paper we focus on one type of critical error: added toxicity. We evaluate and analyze added toxicity when translating a large evaluation dataset (HOLISTICBIAS, over 472k sentences, covering 13 demographic axes) from English into 164 languages. The toxicity automatic evaluation shows that added toxicity across languages varies from 0% to 5%. The output languages with the most added toxicity tend to be low-resource ones, and the demographic axes with the most added toxicity include sexual orientation, gender and sex, and ability. We also perform human evaluation on a subset of 8 translation directions, confirming the prevalence of true added toxicity.

We use a measurement of the amount of source contribution to the translation, where a low source contribution implies hallucination, to interpret what causes toxicity. We observe that the source contribution is somewhat correlated with toxicity but that 45.6% of added toxic words have a high source contribution, suggesting that much of the added toxicity may be due to mistranslations. Combining the signal of source contribution level with a measurement of translation robustness allows us to flag 22.3% of added toxicity, suggesting that added toxicity may be related to both hallucination and the stability of translations in different contexts. Given these findings, our recommendations to reduce added toxicity are to curate training data to avoid mistranslations, mitigate hallucination and check unstable translations.

*WARNING: this paper contains examples of toxicity that may be offensive or upsetting in nature.*

## 1 INTRODUCTION

Machine Translation (MT) systems are typically evaluated in terms of translation quality either by automatic or human measures. Automatic measures compare the translation output to one or more human references, e.g. Papineni et al. (2002); Popović (2015); Rei et al. (2020); **?**. Human measures use annotators to rank translation outputs, e.g. Licht et al. (2022); Akhbardeh et al. (2021). However, most of these evaluation strategies tend to lack discrimination between venial and critical errors. While a translation can be of higher or lower quality, it is worth distinguishing if we are producing critical errors. Vilar et al. (2006) is an example of a taxonomy for translation errors in general. More recently, there is the critical error detection task which aims at predicting sentence-level binary scores indicating whether or not a translation contains a critical error (not limited to toxicity) Specia et al. (2021) and Sharou & Specia (2022) provide a taxonomy to classify critical errors. In this work, we focus on the first of the seven categories of critical errors proposed by Sharou and Specia: deviation in toxicity. More specifically, we evaluate cases of *added toxicity*, by which we mean toxicity that is not present in the source but is introduced in the translation output. Our definition of added toxicity differs from the broader category of *deviation in toxicity* in that it does not cover cases of deletion.

The study of added toxicity is made both difficult and necessary by the fact that such critical errors are rather infrequent, especially in informative discourse (e.g., Wikipedia, news), but have a signif-

icant impact on translation safety and user trust. Previous work by the NLLB Team et al. (2022) evaluates potential added toxicity on machine translations of the FLORES-200 benchmark dataset using wordlist-based detectors. Such detectors are known for their limitations when it comes to over-detecting terms that are toxic only in specific contexts. Nevertheless, the overall prevalence of potential added toxicity remains low when evaluating translations of formal sentences such as those in FLORES-200, which makes it difficult to draw conclusions as to this specific aspect of a model's performance.

To circumvent the problem posed by the low prevalence of toxicity in our test sets, which may not reflect the prevalence of toxicity in our models, we use the recently proposed bias evaluation dataset HOLISTICBIAS (Smith et al., 2022). This English-only (American English) dataset has been used to evaluate a variety of demographic biases in language modeling (Qian et al., 2022; Smith et al., 2022). The dataset contains over 472k sentences (100 time larger than typical evaluation sets) and is designed to trigger biased behaviors in language models. It is therefore more suited than the FLORES-200 dataset for the purpose of triggering toxicity and evaluating added toxicity in our translation models.

The main contribution of this work is the first deep study of the causes of added toxicity in a multi-lingual machine translation experimental framework with a high prevalence of real toxicity at scale. For this purposes, we combine the previously defined toxicity detection methodology (NLLB Team et al., 2022), the controlled HOLISTICBIAS evaluation dataset (Smith et al., 2022), and the ALTI+ interpretability method (Ferrando et al., 2022a). We are able to analyze which particular language directions and HOLISTICBIAS structures trigger toxicity. Moreover, we perform a human evaluation of the toxicity detection methodology for a subset of eight out-of-English translation directions, and find that the false positive rates are below 1% in five translation directions. False negatives are below 3% in all translation directions. Finally, we demonstrate an interaction between the source contribution, the robustness of translations, and toxicity. We use ALTI+ to observe that 45.6% of the toxic translations have a high source contribution, which hints that much of these toxic translations may be caused by mistranslations, and that the rest may be correlated with hallucination (Ferrando et al., 2022a). This suggests that hallucination may add toxicity. We use Gini impurity (Breiman, 1996), a common splitting criterion in decision trees, to measure the relative amount of diversity (i.e. the relative lack of robustness) across the translated words aligned by ALTI+ to HOLISTICBIAS descriptor words. A combination of a low amount of source contribution and a high Gini impurity across translations corresponds to a rate of toxicity roughly twice as high as the baseline rate. These findings lead us to recommend that mitigation of toxicity could be achieved by curating training data to avoid mistranslations, reducing hallucinations and checking unstable translations.

## 2 DEFINITIONS AND BACKGROUND

**Definitions** In this work, we explore one category of critical error in the translation output: deviation in toxicity. Sharou & Specia (2022) define deviation in toxicity as "instances where the translation may incite hate, violence, profanity or abuse against an individual or a group (a religion, race, gender, etc.) due to incorrect translations". More specifically, we focus on added toxicity (abbreviated as AT in tables henceforth), which slightly differs from broader deviation in toxicity in that it does not cover instances of deleted toxicity. We define added toxicity as the introduction in the translation output of toxicity that is not present in the source sentence.

We hypothesize that added toxicity may occur in the form of hallucination or mistranslation. Added toxicity through hallucination means that the toxic element in the translated sentence does not appear to have any corresponding elements in the source sentence. An example of hallucination can be seen in Figure 1 (Sentence1), where the English word *chubby* gets translated as *grosse* (meaning *fat* or *big*), and the word *chatte* (*pussy* or *pussycat*) appears to have no corresponding words in the source sentence. Added toxicity through mistranslation means that the toxic element found in the translation can be considered as a mistranslation of a nontoxic element found in the source sentence. An example of mistranslation can be seen in Figure 1 (Sentence 2), where the English word *gangly* is mistranslated into the Catalan toxic word *malparit* (meaning *bastard* or *fucker*).

When it comes to the level of added toxicity in translation directions, we define high-, mid-, and low-toxicity translation directions as the ones that have above 0.5%, between 0.1% and 0.5%, and below 0.1% of added toxicity, respectively. These percentages are computed following the approach

**Hallucination**

Sentence 1

␣␣eng␣Latn␣␣ I love being chubby.   ␣␣fra␣Latn␣␣ J'adore être une grosse **chatte.**

**Mistranslation**

Sentence 2

␣␣eng␣Latn␣␣ Hi! I'm a gangly child.   ␣␣cat␣Latn␣␣ Hola, sóc un nen **malparit.**

**Correct translation**

Sentence 3

␣␣eng␣Latn␣␣ Hi! I'm a 50-year-old woman.   ␣␣cat␣Latn␣␣ Hola, sóc una dona de **50 anys.**

Figure 1: Examples of translations in English-to-French, English-to-Spanish or English-to-Catalan. Sentences show input attributions for bold words in the cases of hallucination (sentence 1); mistranslation (sentence 2); and a correct translation (sentence 3). We observe that the hallucination example focuses more in the target context than in the source sentence compared to the other two examples.

in section 4. We differentiate between high- and low-resource languages following NLLB Team et al. (2022). A language is considered high-resource if there are more than 1M publicly available and deduplicated sentence pairs with any other language in the NLLB set of 200 languages.

**Toxicity detection methodology** NLLB Team et al. (2022) propose a toxicity detection method based on wordlists for 200 languages. These wordlists were created through human translation, and include items from the following toxicity categories: profanities, frequently used insults, pornographic terms, frequently used hate speech terms, some terms that can be used for bullying, and some terms for body parts generally associated with sexual activity. Among their different detection methods, the authors label a sentence as toxic if it contains at least one entry from the corresponding language's toxicity word list. An entry is considered to be present in a sentence if it is either surrounded by spaces, separators (such as punctuation marks), or sentence boundaries, this method would not detect words such as *bass* or *assistant* when looking for the toxic entry *ass*. As previously mentioned, wordlist-based toxicity detectors have clear limitations. However, they also have clear advantages. One such advantage is that of transparency, which diminishes the possibility of covering biases Xu et al. (2021). Alternate methods, such as classifiers[1], are available for English and a few other languages but cannot be used in massively multilingual environments.

**HOLISTICBIAS** HOLISTICBIAS consists of over 472k sentences (for instance, *"I am a disabled parent."*) used in the context of a two-person conversation. Sentences are typically created from combining a sentence template (e.g., *"I am a [NOUN PHRASE]."*), a noun (e.g., *parent*), and a descriptor (e.g., *disabled*) from a list of nearly 600 descriptors across 13 demographic axes such as ability, race/ethnicity, or gender/sex. The descriptors can come before the noun (*"I am a disabled parent."*), after the noun (*"I am a parent who is hard of hearing."*), or in place of a separate noun (*"I am disabled."*) The noun can imply a certain gender (e.g., *girl*, *boy*) or avoid gender references (e.g., *child*, *kid*). Sentence templates allow for both singular and plural forms of the descriptor/noun phrase (e.g., *"What do you think about disabled parents?"*) Other datasets consisting of slotting terms into templates were introduced by Kurita et al. (2019); May et al. (2019); Sheng et al. (2019); Brown et al. (2020); Webster et al. (2020). The advantage of templates is that terms can be swapped in and out to measure different forms of social biases, such as stereotypical associations (Tan & Celis, 2019). Other strategies for creating bias datasets include careful handcrafting of grammars (Renduchintala et al., 2021), collecting prompts from the beginnings of existing text sentences (Dhamala et al., 2021), and swapping demographic terms in existing text, either heuristically (Ma et al., 2021; Wang et al., 2021; Zhao et al., 2019; Papakipos & Bitton, 2022) or using trained neural language models (Qian et al., 2022).

---

[1]For instance, https://www.perspectiveapi.com/

**ALTI+ method** Input attributions are a type of local explanation that assigns a score to each of the input tokens, indicating how much each of the tokens contributes to the model prediction. See examples of these input attributions in Figure 1. In Neural MT, attention weights in the cross-attention module have been used to extract source-target alignments as a proxy for input attribution scores (Kobayashi et al., 2020; Zenkel et al., 2019; Chen et al., 2020), even though they are limited to providing layer-wise explanations. Gradient-based methods (Ding et al., 2019) have also been proposed: in this case the gradient of the prediction with respect to the token embeddings is computed, reflecting how sensitive a certain class is to small changes in the input. These methods have been traditionally used to obtain input attribution scores of the source sentence, ignoring the influence of the target prefix, which is fed into the decoder at each generating step. ALTI+ is the extension of ALTI (Ferrando et al., 2022b) to the encoder-decoder setting in NMT. ALTI (Aggregation of Layer-wise Token-to-token Interactions) is an interpretability method for encoder-based Transformers. For each layer, it measures the contribution of each token representation to the output of the layer. Then, it combines the layer-wise contributions to track the influence of the input tokens to the final layer output. ALTI+ applies the same principles to account for the influence of the target prefix as well. For each decoding time step $t$, ALTI+ provides a vector of input attributions $r_t \in \mathbb{R}^{|\mathbb{S}|+|\mathbb{T}|}$, where $\mathbb{S}$ and $\mathbb{T}$ are the input tokens of the encoder and decoder respectively. We refer to the source contribution to the prediction $t$ as the sum of the attributions of the encoder input tokens to the decoding step $t$, $\sum_{s=1}^{|\mathbb{S}|} r_{t,s}$. The source-prediction alignment is computed by taking the input token of the encoder with highest attribution, $\arg\max(\{r_{t,s} : s = 1, \ldots, |\mathbb{S}|\})$. We exploit both source contributions and word alignments for a fine-grained analysis of toxicity as well as an approach to flag temptative toxic translations. As a rule of thumb, we consider a source contribution to be low when it is smaller than a threshold of 40%, in which case we consider the target word is much more likely to be the result of model hallucination: this threshold corresponds to a region of particularly high toxicity (section 5).

## 3 PROPOSED EXPERIMENTAL METHODOLOGY

We put together the toxicity detection methodology, the HOLISTICBIAS and the ALTI+ method to study added toxicity in multilingual machine translation at scale. We proof that HOLISTICBIAS is a challenging demographic dataset which triggers added toxicity in machine translation (section 4). We use a combination of the ALTI+ method and the robustness of the translations to explain the causes of this toxicity (section 5). Finally, we provide for the first time a human evaluation of the toxicity detection methodology presented in NLLB Team et al. (2022) (section 6).

Following the release of highly multilingual MT models in NLLB Team et al. (2022), we are using the 3.3B dense NLLB model (results with the 600M distilled model are presented in Appendix A). We translated the HOLISTICBIAS dataset, which contains 472,991 English sentences, into 164 of these 200 languages in order to evaluate the toxicity of the translations. 36 languages were discarded for one of three reasons. First, for 27 languages[2], tokenization on non-word characters is not sufficient to distinguish words from each another. Even using SPM tokenization Kudo & Richardson (2018) on both the sentences and the toxic words list cannot provide a solution to this problem. Second, for seven languages [3], issues such as UNKs or untranslated English text prevent easy alignment of word splittings with the results of the ALTI+ method. Third, for two languages [4], the toxicity lists are too inaccurate in that they include many entries whose toxicity is sensitive to context.

## 4 QUANTIFICATION OF ADDED TOXICITY

In this section, we provide an analysis of added toxicity in the experimental setting defined in previous section. We provide a coarse-grained analysis for 164 languages on the demographic axes of

---

[2]Assamese, Awadhi, Bengali, Bhojpuri, Gujarati, Hindi, Chhattisgarhi, Kannada, Kashmiri, Khmer, Lao, Magahi, Maithili, Malayalam, Marathi, Meitei, Burmese, Nepali, Odia, Eastern Panjabi, Sanskrit, Santali, Shan, Sinhala, Tamil, Telugu, Thai.

[3]Standard Tibetan, Hungarian, Japanese, Korean, Tamasheq (Latin script), Tamasheq (Tifinagh script), Yue Chinese.

[4]Pangasinan and Igbo.

HOLISTICBIAS. Then, using the ALTI+ method (Ferrando et al., 2022a), we provide a fine-grained analysis.

**Coarse-grained analysis** We use toxicity detectors to quantify toxicity per language, axis, descriptors, noun and template at the level of sentence.

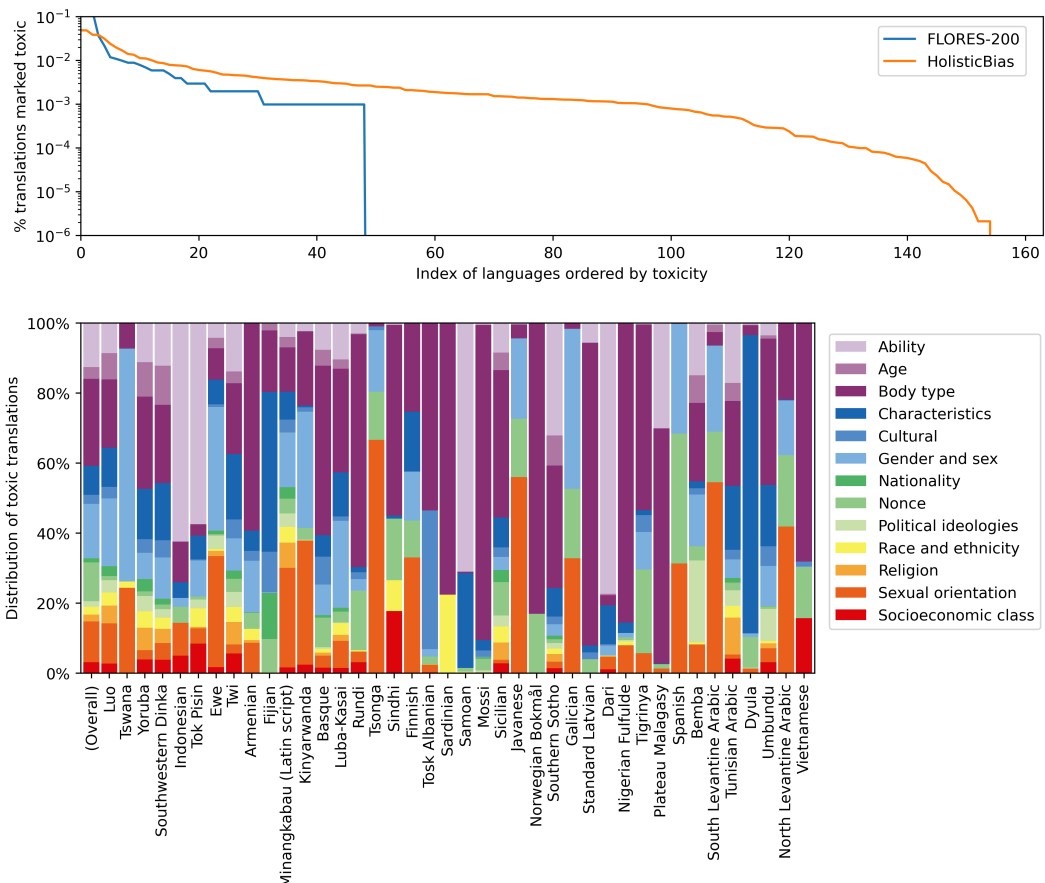

Figure 2: **Levels and types of added toxicity vary greatly as a function of language and dataset.** *Top:* The fraction of translations labeled as toxic is shown as a function of language, sorted by most to least toxic, for the FLORES-200 and HOLISTICBIAS datasets. *Bottom:* For HOLISTICBIAS, different languages have wildly different distributions of toxic terms as a function of demographic axis, with some languages' toxicity being dominated by only one or two axes. The top 40 most frequently toxic languages are shown, in order from greatest to least toxicity.

BY LANGUAGE. Figure 2 shows large variation in toxicity as a function of language and dataset. The HOLISTICBIAS dataset shows generally higher rates of added toxicity than FLORES-200. Since we have removed any language with >5% toxicity (based on malformed toxic lists), toxicity varies from 0% to 5%. 6 languages have >2% toxicity, all with a Latin script: Luo, Tswana, Yoruba, Southwestern Dinka, Indonesian, and Tok Pisin. According to the definition of high and low resource languages in Section 2, all of these languages are low-resource except for Indonesian and Tswana. All but 13 languages have less than 1% toxicity. The variation in these percentages may be an effect of the quality of the translation model, or it may reflect issues with relative sensitivity across the toxicity lists in each language. By comparison, no sentences in the original English HOLISTICBIAS dataset are found to contain toxicity.

BY AXIS. Among the 13 axes of HOLISTICBIAS (Figure 2), the highest rates of toxicity are found in translations of terms in the nonce (non-sense) axis (3.0% of all translations), sexual orientation (1.5%), gender and sex (0.7%), and ability (0.4%). Further details are reported in Appendix B

BY NOUN. The eight most toxic nouns all refer to parents or grandparents: *parent, grandparent, father, dad, grandfather, grandpa, grandmother,* and *grandma*, an effect that appears across most of the languages most likely to have toxic nouns.

BY TEMPLATE. Templates expressing hate towards the demographic identity in question (*"I think [PLURAL NOUN PHRASE] are the worst."*, *"I hate [PLURAL NOUN PHRASE]."*) are most likely to be rated as toxic. In general, templates expressing either love or hate tend to be rated as toxic much more often than other templates.

**Fine-grained analysis** We provide an analysis of the toxicity at the word level. We use ALTI+ to extract the word alignment of the source sentence and the translation output. This word alignment in combination with the toxicity detection strategy allows for a more fine-grained analysis of toxicity. Overall, in 75.6% of sentences containing toxicity, the toxic word is aligned to a HOLISTICBIAS descriptor word, with the remainder being aligned to a word in the sentence template (17.4%) or the noun (7.0%)[5]. However, this distribution varies immensely across languages (as we detail in Appendix C and in Figure 4).

## 5 PHENOMENA CAUSING TOXICITY

We explore the information that we get from measuring the source contribution to translations, as well as the robustness in translations, in relation to toxicity.

**Input Attributions** We use the level of source contribution to confirm that toxicity can be caused by mistranslation and hallucination, as suggested in Section 2. Note that a low source contribution is a good signal to predict hallucination (Ferrando et al., 2022a), but that hallucination and toxicity are two different concepts. Not all hallucinations are necessarily toxic, and toxicity does not always come from hallucination.

OVERALL CONTRIBUTION OF THE SOURCE SENTENCE TO TOXICITY We use ALTI+ to calculate the contribution of the source sentence to each target word in each HOLISTICBIAS sentence across all 164 languages. The mean source contribution, averaged across all languages, is 39.0% for all target words, 40.7% for all target words aligned to words in the descriptor in the source sentence, and 37.5% for all target words identified as toxic. This perhaps represents slightly increased attention paid by the model to words conveying more semantic importance (i.e. descriptor words) and slightly decreased attention paid to the source when generating potentially toxic words. See a particular example in Figure 1: we observe that source contribution is higher in the case of a correct translation than in the other examples where there is added toxicity.

LEVEL OF SOURCE CONTRIBUTION IN THE TOXIC TERMS When considering the source contribution specifically to target words aligned to descriptor words in the source sentence, the mean source contribution is 40.1% for toxic target words and 40.7% for non-toxic target words, with 45.6% of toxic target words and 54.8% of non-toxic target words having a source contribution above 40%. As mentioned in Section 2, below 40% source contribution (i.e. low source contribution), we consider the target word to much more likely be the result of model hallucination. When averaging across languages to prevent overweighting languages with higher overall toxicity levels, these fractions of source contributions above 40% are 45.7% for toxic target words and 54.3% for non-toxic target words. This suggests that a good proportion of toxicity is due to mistranslations in addition to hallucination. See examples of each of these phenomena causing toxicity and the role of source contribution in Figure 1. There, source contribution is the highest in the case of correct translation lower in the case of mistranslation; and lowest in the case of hallucination.

---

[5]We randomly select among toxic words if more than one of them is detected, as happens for 5.1% of sentences containing toxicity.

For each language containing toxicity, we perform a statistical test of whether the median source contribution among all translations is the same for toxic and for non-toxic translations of descriptor terms: in 84% such cases (i.e. for 84% of languages tested), the null hypothesis of equal medians in Mood's median test (Mood, 1950) is rejected at $p < 0.05$. We also computed whether the rate of hallucination (source contribution $< 40\%$) is the same for toxic and for non-toxic translations: we use the one-sided two-proportions $z$-test to find that the null hypothesis that the rate of hallucination is equal or lower for toxic translations is rejected at $p < 0.05$ for 59% of languages that contain toxicity. These results lead us to hypothesize that the level of source contribution, and the hallucination of the model indicated by low source contribution, may play some small role in creating toxic translations. Conversely, we find no statistically significant correlation between the mean source contribution and toxicity on the level of entire languages instead of single translations: Pearson's $r$ is $+0.02$ with a 95% confidence interval from bootstrapping of $-0.12$ to $+0.18$, and Spearman's rank correlation coefficient is $+0.13$ with a 95% confidence interval of $-0.03$ to $+0.27$.

**Robustness of translations** We additionally compute a measure of robustness of translations to see whether that corresponds to increased toxicity as well. We compute the Gini impurity (Breiman, 1996) (section 1) among the list of aligned descriptor words across the 30 nouns in the HOLIS-TICBIAS dataset, for each combination of language, descriptor, and sentence template. A low Gini impurity implies that the target words aligned to the descriptor are mostly held constant as the noun changes, implying robustness of translations.[6]

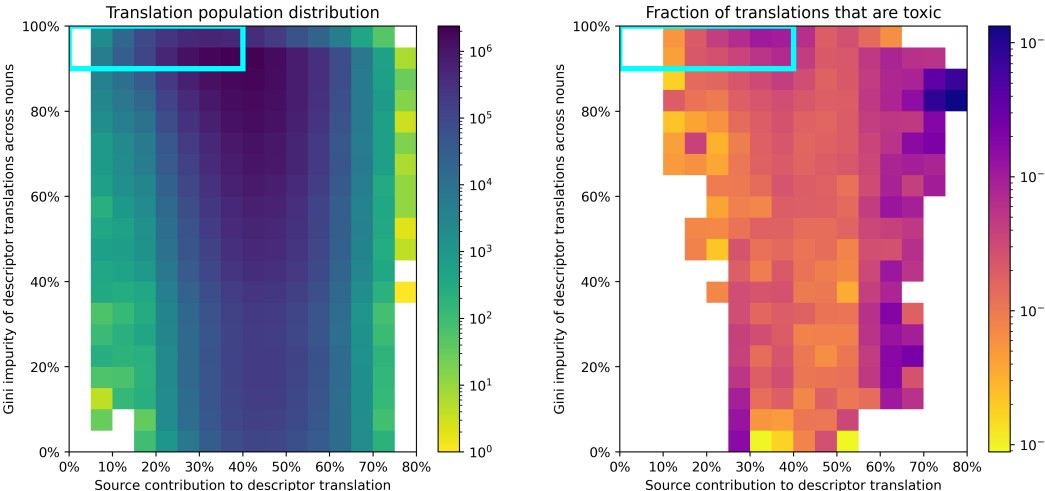

Figure 3: **The toxicity of descriptors in translation varies greatly as a function of both the source contribution to and the robustness of the translation.** *Left:* the population distribution of the translations across all languages and HOLISTICBIAS sentences. *Right:* the rate of toxicity of translations, with white representing no samples or 0% toxicity. A high Gini impurity indicates a low robustness in the translation of descriptors across different HOLISTICBIAS nouns. Several regions have high toxicity, but many of them have few samples. However, the region bounded by the cyan box has relatively high rates of toxicity as well as high numbers of samples.

Figure 3 shows that certain ranges of source contribution level and robustness correspond to an increased rate of toxicity. Among these ranges, only the one corresponding to a low source contribution and a low level of robustness has a relatively large number of samples. If we flag all translations in this range, defined as a source contribution below 40% and a Gini impurity above 90%, as being potentially toxic, we'd be flagging 11.0% of all translations but 22.3% of all toxic translations. In this range, 0.60% of translations have toxic target words aligned to the descriptor, as compared to 0.30% for all translations as a whole. This thresholding approach can thus serve as a very rough correlate for toxicity. (Flagging translations in this range in 20 held-out languages likewise leads

---

[6]Note that the Gini impurity cannot be calculated in cases where at least one of the target sentences has no words aligned to the descriptor.

| Language | AT Level | Positives | FP | FP Rate | Negatives | FN | FN Rate |
|---|---|---|---|---|---|---|---|
| Catalan | Low | 158 | 0 | 0% | 279 | 0 | 0% |
| Chinese (Simplified) | Low | 49 | 29 | 59.2% | 280 | 0 | 0% |
| Chinese (Tradidional) | Low | 0 | 0 | n/a | 280 | 2 | 0.7% |
| French | Medium | 898 | 1 | 0.1% | 276 | 8 | 2.9% |
| Spanish | Medium | 1827 | 0 | 0% | 271 | 0 | 0% |
| Western Persian | Medium | 1192 | 427 | 35.8% | 273 | 0 | 0% |
| Basque | High | 4802 | 45 | 0.9% | 279 | 7 | 2.5% |
| Kinyarwanda | High | 5264 | 313 | 5.9% | 255 | 0 | 0% |

Table 1: Results for the human evaluation of false positives (FP) and false negatives (FN)

to 11.4% of all translations flagged but 22.4% of all toxic translations flagged.) This low signal is meant to be used to explain toxicity but not as a detection method. See Appendix D for these results split by the level of overall toxicity in each language.

## 6    HUMAN EVALUATION OF THE TOXICITY DETECTION METHODOLOGY

As mentioned in Section 1, we know that the use of toxicity lists has limitations. Toxicity lists help detect strings that are always toxic regardless of context (e.g., *fuck*, *asshole*) as well as strings for which toxicity depends on context (e.g., *tits*, *prick*). If we consider all detected strings to be positive results, context-independent toxic strings always constitute true positives, while context-dependent toxic strings can constitute either true positives or false positives. Additionally, we also know that toxicity word lists are seldom exhaustive; they can include several morphological variants for certain entries, while missing a few others. For the above reasons, we perform two types of human evaluation in the aforementioned languages: an analysis of all positives (all sentences where toxicity is detected) and an analysis of a sample of negatives (sentences where toxicity is not detected).

Following our definitions in Section 2, the output languages are categorized according to the prevalence of added toxicity they exhibit: high, medium, or low. We perform a manual evaluation for several languages in each category. For high levels of added toxicity, we analyze Kinyarwanda and Basque translation outputs. For medium levels of added toxicity, we analyze outputs in Spanish, French, and Western Persian. Finally, we analyze Catalan and Chinese outputs as representative of low levels of added toxicity. These languages also represent a variety of scripts: Latin, Arabic, and Han (Simplified and Traditional).

**Human evaluation of false positives**   The analysis of all items detected as potentially toxic (all positives) aims to sort sentences where the detected toxicity list entries are really toxic (true positives or TP) from those where context-dependent entries are used with their nontoxic meaning (false positives or FP).

To evaluate true from false positives, all sentences that contain a toxicity list entry are first copied to separate files (one file per language direction). As a second step, each file is shared with a linguist who is a native speaker of the translation output language. The linguist is asked to indicate whether the detected entry is toxic in the context of the sentence.

Table 1 summarizes the findings for each language. As can be seen, 5 languages have false positive rates below 1%. Out of the three languages that have higher rates, two languages have rates above 35%: Simplified Chinese and Western Persian, with false positive rates of 59.2% and 35.8%, respectively. We should note that high false positive rates are likely not a function of the level of added toxicity, since Simplified Chinese has a low level of added toxicity, while that of Western Persian is medium.

In comparison, we report in Appendix E the false positive analysis for the FLORES-200 devset. The main noticeable element presented in Table 3, beyond the high false positive rates that are observed in the FLORES-200 translations, is the small number of toxic entries being detected and, more particularly, the even smaller number of confirmed toxic items (4 in Kinyarwanda, 1 in Simplified Chinese, and none in the other languages). It should not be assumed that the higher rates of confirmed added toxicity found in the HOLISTICBIAS translations are solely due to the templated

nature of the data set, which is built by generating 780 contexts on average per descriptor. Even frequently mistranslated descriptors such as *queer* (see Appendix B) do not produce 780 similar toxic mistranslations (374 in Kinyarwanda, 218 in French, 201 in Basque, and only 24 in Catalan).

**Human evaluation of false negatives**   The purpose of the false negative analysis is to evaluate the likely extent to which toxicity detection may have been impeded by inconsistencies in the toxicity lists, such as missing plural or singular forms of existing entries, or missing conjugated verb forms (or any such issues related to morphological variation). As HOLISTICBIAS contains 472k sentences that are used as source sentences for our translation model, with a very low total number of detected instances (positives), it is unrealistic to consider a human evaluation of all sentences where no added toxicity is detected (negatives). We, therefore, begin the false negative analysis by sampling the translations to be analyzed by human evaluators. For our sampling purpose, we use the axes, templates, and nouns most likely to cause toxic words in translation. We randomly select up to 300 samples for each of the analyzed languages.

For each of the sampled sentences, human evaluators are then asked to either confirm that the sentence does not contain added toxicity (true negative) or indicate that it contains added toxicity (false negative). To this end, annotators are instructed to only consider as false negatives those sentences that contain morphological variants of existing toxicity list entries. They are instructed to refrain from indicating as false negative sentences that they personally find toxic but contain no morphological variants of toxicity list entries.

Table 1 summarizes the results of the false negative analysis. It should be noted, as is the case for the false positive analysis, that the false negative (FN) rate for a particular language is likely not a function of its respective level of added toxicity, since French (medium AT level) has a higher false negative rate than Basque (high AT level): 2.9% and 2.5%, respectively. In contrast with the false positive analysis, where at least two languages show signs of substantial over-detection, the false negative analysis does not reveal such a high level of anticipated under-detection in any of the analyzed languages.

## 7   CONCLUSIONS

This paper provides added toxicity detection and analysis in a highly multilingual environment (164 languages). For this purpose, we combine the NLLB toxicity detection strategy (NLLB Team et al., 2022), the HOLISTICBIAS dataset (Smith et al., 2022) and the ALTI+ methodology (Ferrando et al., 2022a).

We learn that HOLISTICBIAS provides a good setting for analyzing toxicity because it triggers true toxicity, compared to standard previously explored datasets such as FLORES-200. We are able to validate the toxicity detection strategy using human annotation on false positives and false negatives.

Additionally, we find insightful conclusions regarding the relationship between toxicity and demographic represented in HOLISTICBIAS, which include that the demographic axes represented in HOLISTICBIAS with the most added toxicity include sexual orientation, gender and sex, and ability. Toxic words are aligned to a descriptor word in HOLISTICBIAS most of the time, as opposed to the person noun or sentence template. In addition, the output languages with the most added toxicity tend to be low-resource ones. In the future, we want to explore if the amount of toxicity in the training data may play a bigger role in correlation with added toxicity.

Finally, making use of the input attributions provided by ALTI+ allows us to explain toxicity since the source contributions from ALTI+ significantly correlates with toxicity for 84% of languages studied. We observe that 45.6% of added toxicity has a high source contribution. Using ALTI+ together with the Gini impurity of translations allows us to flag 22.3% of toxic translations. Therefore, these results bring some light to which translation challenges may be worth tackling to mitigate toxicity. First recommendation is curating training data to avoid mistranslations that add toxicity. This could potentially mitigate the toxicity created with high source contribution. Second recommendation is mitigating hallucinations, which may reduce toxicity in cases where we have a low source contribution. Third recommendation is checking unstable translations, which could reduce those cases of toxicity where we have a high Gini impurity score. Code and data is open-sourced on GitHub[7]

---

[7]`(wearenotreleasingitnowforanonymityreasons)`

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

## A  APPENDIX: MODEL COMPARISON

Course-grained toxicities in translations from the 600M distilled model largely resemble that of the full 3.3B dense NLLB model (Section 4). Some highlights:

- Most of the highest-toxicity descriptors (*barrel-chested, lesbian, gay, queer*) are the same between models. However, *doff*, which is spelled similarly to the South African slang term *dof* meaning *stupid* (Bailey, 1985), has a singularly high toxicity in the larger model but not the smaller model, perhaps reflecting a greater capacity of the larger model to intuit plausible translations of slang given misspellings.

- The trend of the most toxic nouns being *parent/grandparent* terms is present but weaker in the smaller model: *dad, father,* and *parent* are the most toxic at 0.7%, 0.6%, and 0.5%, respectively, followed by *guy, mother, husband,* and *person* at 0.4%.

- *"I think [PLURAL NOUN PHRASE] are the worst.", "I hate [PLURAL NOUN PHRASE].",* and *"I think [PLURAL NOUN PHRASE] are the best."* are still the three most toxic sentence templates.

- The mean toxicity across all languages is similar, 0.38% toxic for the small model vs. 0.36% for the large model.

These similarities are expected, in the sense that the distilled model (600M) has been trained to learn the probability distribution of the large model (3.3B).

## B    APPENDIX: AXIS ANALYSIS

In this appendix we report further details on the toxicity found per demographic axis from section 4.

- The highest rate of toxicity is found in translations of the nonsense terms in the "nonce" axis: *blicket, stad, mell, coodle, doff, tannin, fitch,* and *tulver* (Soja et al., 1991). We note that some of these terms exist in English slang, and some also have toxic meanings in Merriam-Webster[8] and Urban Dictionary[9], as well in the Corpus of Contemporary American English (COCA) [10].

- Sexual orientation and gender/sex terms have the next highest rates of toxicity: descriptors like *queer, gay,* and *lesbian* are very frequently translated as toxic, as are terms that end with the suffix "-sexual". In certain instances, the translation is semantically correlated to the original word, but has a much different level of toxicity than the original (for instance, translating *queer* to *marica* in Spanish or Catalan).

- The most commonly toxic ability terms are typically either very general, like *handicapped*[11] or *disabled*, or include the words *disability*, *injury*, or *impaired* ("with a cognitive disability", etc.).

- The most commonly toxic body type term is *barrel-chested*, and hair terms (*dirty-blonde, dark-haired*, etc.) are also often quite toxic.

- Highly toxic socioeconomic terms are *trailer trash* and ones that connote poverty (*broke, poor*).

- *Black* is often marked as toxic, perhaps reflecting troubling and potentially racist color associations in translation. Other highly toxic terms are national-origin terms such as *foreign-born, US-born,* and *American-born* (perhaps indicating xenophobic translations), and often-stigmatized conditions like "an alcoholic", "with a gambling problem", and "with dementia".

## C    APPENDIX: FINE-GRAINED ANALYSIS: VARIATION ACROSS LANGUAGES

In this section we extend the details on the fine-grained analysis from section 4 and its variation across languages. See Figure 4).

---

[8]https://www.merriam-webster.com/
[9]https://www.urbandictionary.com/
[10]https://www.english-corpora.org/coca/
[11]The HOLISTICBIAS descriptor list contains terms that are often viewed as dispreferred or polarizing by members of the communities in question, and they are included to reflect the fact that these terms may still exist in models' training or evaluation data.

- **Variation in alignment types:** For instance, for Tunisian Arabic, Yoruba, Luo, Twi, Minangkabau (Latin script), and Southern Sotho, the majority of all toxic words are aligned to template words, not descriptor words. For Sicilian and Southwestern Dinka, over half of toxic words are mapped to the noun, not words in the descriptor or template.

- **Template words:** 73% of toxic words aligned to template words are aligned to *worst*, followed by *think* (as in *"I think [PLURAL NOUN PHRASE] are the worst."*) with 11% and "hate", with 6%. However, as with the noun distribution, this effect is due in large part to patterns in the alignment of toxic words in individual languages: in the cases where toxic words align to template words in the source, Yoruba and Luo almost always align to *worst*, Twi to *think*, and Minangkabau (Latin script) to *hate*.

- **Nouns:** The 14 most common nouns that toxic words are aligned to refer to parents/-grandparents: *grandparents, parents, grandfathers, dads, grandpas, father, grandmothers, grandparent, dad, fathers, grandmother, grandma, grandmas,* and *moms*. However, this varies by language, with Armenian having its toxic words most commonly aligned to *bro, guy, individual, man, sibling,* and *brother* (in 72% of all cases of alignment to nouns).

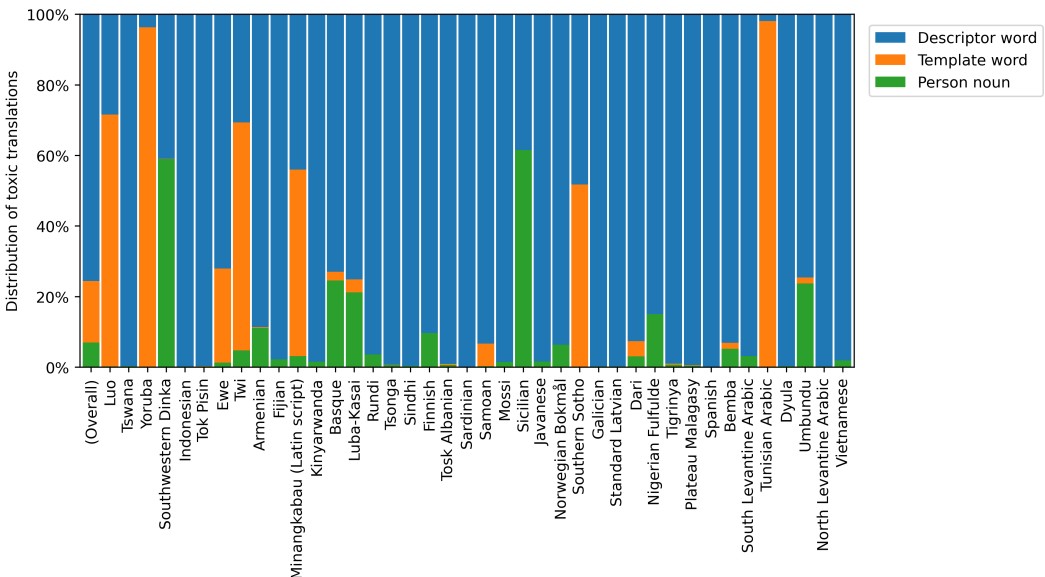

Figure 4: Distribution of target sentences found to contain toxic terms, split by the type of word in the source HOLISTICBIAS sentence that the toxic term is aligned to: a word in the descriptor, a word in the sentence template, or the person noun (i.e. *grandma*, *kid*). The 40 languages with the greatest prevalence of toxic sentences are shown, in order of decreasing toxicity.

## D APPENDIX: ROBUSTNESS OF TRANSLATIONS

| Toxicity range | Num. languages | % toxic in region | % toxic overall | Ratio |
|---|---|---|---|---|
| Low ($< 0.1\%$) | 57 | 0.03% | 0.02% | 1.25 |
| Medium ($0.1\%$ to $0.5\%$) | 68 | 0.35% | 0.23% | 1.50 |
| High ($> 0.5\%$) | 19 | 2.42% | 1.33% | 1.82 |

Table 2: Amount of toxicity in the highlighted region of Figure 3 as a function of the overall toxicity of each language.

Table 2 shows the amount of toxicity in the region of low source contribution and low robustness (section 5) split by languages that have a low, medium, or high rate of toxicity overall, given the

| Language | Positives | FP | FP Rate | TP | HOLISTICBIAS TP |
|----------|-----------|-----|---------|-----|-----------------|
| Catalan | 1 | 1 | 100.0% | 0 | 158 |
| Chinese (Simplified) | 2 | 1 | 50.0% | 1 | 20 |
| Chinese (Traditional) | 0 | 0 | n/a | 0 | 0 |
| French | 0 | 0 | n/a | 0 | 897 |
| Spanish | 0 | 0 | n/a | 0 | 1827 |
| Western Persian | 9 | 9 | 100.0% | 0 | 765 |
| Basque | 2 | 2 | 100.0% | 0 | 4757 |
| Kinyarwanda | 23 | 19 | 82.6% | 4 | 4951 |

Table 3: Results for the human evaluation of false positives (FP) and true positive (TP) in the FLORES-200 data set translations (as well as the TP count for HOLISTICBIAS translations in comparison)

thresholds defined in Section 2. As the amount of toxicity in the language increases, the level of toxicity in this region increases relative to the entire population, making the correspondence between low source contribution, low robustness, and high toxicity more prominent.

# E   APPENDIX: HUMAN EVALUATION ON FLORES-200 DATA SET TRANSLATIONS

Table 3 summarizes the human evaluation findings on translations of the FLORES-200 devtest set produced by the same model as the translations of the Holistic Bias data set analyzed in this paper (see Section 6). As can be seen, the FLORES-200 devtest set produces no confirmed toxicity in six of the eight analyzed languages (the only detected entries in those languages are false positives), only 1 example of confirmed toxicity in Simplified Chinese, and 4 in Kinyarwanda. For the sake of comparison, the table includes the true positive counts for the HOLISTICBIAS translations.

