# OpenReview forum: "Toxicity in Multilingual Machine Translation at Scale"
_ICLR.cc/2023/Conference — Submitted to ICLR 2023_

### Official Review · Reviewer_HKPu · 2022-10-21

**Confidence:** 4
**Correctness:** 3
**Technical Novelty And Significance:** 2
**Empirical Novelty And Significance:** 4
**Recommendation:** 8

**Clarity, Quality, Novelty And Reproducibility:**

The paper is very clear except for the problems mentioned above.

I’m not sure what is meant by quality of work, but this seems to be solid science. The persistent use of various statistical tests, and brief mentions of what results one might expect by chance when doing tests over so many languages was a very nice touch.

The work is original enough - it would be easy enough to characterize this as a simple addition of two ideas (list based analysis + an existing template-based corpus), but as I mentioned above, I think this strikes the right balance of interpretability and recall. This is a domain where one doesn’t necessarily want to be clever or fancy at the cost of being actionable and scalable. I worry that much of 3.3+3.4 is an attempt to add some technical weight to the paper, where I think more analysis of the actual text might have been more useful and interesting.

**Strength And Weaknesses:**

Strengths:

This is an important problem, and it’s nice to have a paper dedicated to it.

The paper is mostly clear and easy to follow.

The proposed list-based method with synthetic but targeted source sentences seems to strike a good balance of interpretability and recall of actual problems. The included human check goes a long way toward validating this approach. It is likely to be adopted by others.

The characterization of sentences that have added toxicity (plus the details in the appendix) are nice sources of insight and inspiration.

Weaknesses:

I think the paper’s primary contribution (the adoption of the Holistic Bias corpus as a set of source sentences) would have been stronger if they had compared the discovered toxicity rates to those of FLORES-200. It feels like you have to read both papers in order to follow the first and most important sentence of their conclusion.

I didn’t get much out of the Phenomena Causing Toxicity section or the Robustness of Translations section. This felt somewhat out of place compared with the rest of the paper. It was hard to follow, with statistics and methods being introduced and tested in rapid fire, and I’m not sure what was accomplished. I don’t think the goal was to build a glass-box toxicity detector. It would be hard to beat the black-box detector that uses the same word-list based methodology as the gold standard. Instead, the goal appears to have been to explain where toxicity came from in the model. To that end, we are left with the intuition that it is some mix of hallucination and mistranslation of the source. I think this section would have hit a lot harder if the authors had acted on one of their three suggestions (clean mistranslated data, avoid hallucination, be weary of instability) as a way of validating this approach to explanation.

It’s also strange that there is a large section on explanation / detection, and another section on characterizing based on input attributes, but the authors never contrast the two. Which is more predictive of added toxicity? Input characteristics or glass-box features?

I would have liked to have seen more discussion of the False Positive human evaluation. I couldn’t believe it when that section ended! What caused all those False Positives? Could they be fixed by removing one or two problematic list entries, or was it more complicated than that? IE: do these languages inherently resist the list-based approach?

I would have also liked to have seen more discussion (and numbers) regarding the types of sentences leading to added toxicity. As suggested in my summary, it feels like we have the take the authors’ descriptions at face value, with few backing numbers.

There are some editing mistakes that detract from an otherwise clean paper:

The last few sentences of the second paragraph of the introduction are repeated.

The first sentence of the third paragraph of 4.1 talks about “toxicity rates” when I think it means to discuss “false positive rates”.

**Summary Of The Paper:**

This paper describes a method for characterizing the amount of added toxicity in a multilingual machine translation system’s output. The toxicity detection method is a list-based approach that is unchanged from an earlier publication (NLLB Team et al 2022, cited prominently and repeatedly). The main contributions here are:

(1) Moving from FLORES as the test set to translations from English of the recently proposed Holistic Bias corpus. The main insight seems to be that for toxicity analysis, one doesn’t really need references so much as source sentences that cover topics that are likely to elicit the introduction of toxicity.

(2) They provide an analysis of properties of sentences that tend to elicit added toxicity in Section 3.1 Outside of the bottom half of figure 2, this is mostly descriptive. We get the flavor of what matters with little quantification (“most of the languages”, “much more often”).

(3) They use explainability methods plus robustness characterizations to try to explain added toxicity, and to propose a very weak toxicity detector.

(4) They provide a human verification of the toxicity detector’s output, characterizing the False Positive rate by checking each toxic addition, and using insights from (2) to aim annotators toward examples likely to have False Negatives.

**Summary Of The Review:**

This is not a perfect paper, but it’s solid work on an important topic. I suspect that the solution landed on here (word lists + a template-based corpus) will be good enough for most industrial labs; i.e.: this has a shot at being the definitive citation for the problem of detecting and characterizing added toxicity in MT. I would have liked to have seen more input-related numbers and visualizations over broad text-based descriptions, and I wasn’t crazy about the explainability portion, but I don’t think that should hold the paper back from the public.

---

> ### Author Response · Authors · 2022-11-16
> **Added comparison between Holistic Bias and Flores data sets, Extended Human Analysis, and Clarified the description of the Phenomena Causing Toxicity**
>
> We have now included in the new version of the paper the added toxicity with Flores devset. Figure 2 and the human evaluation now include the analysis with this set. We show that added toxicity is lower in the Flores devset (Table 1). Also we show with human analysis that the true positives are much lower in Flores than in HolisticBias (Appendix E).  Additionally, since we have open-sourced our data and findings, the reader can see all examples of toxicity and the types of sentences leading to added toxicity in the corresponding Github  (we are not releasing it now for anonymity reasons)
>
> We have now revised the Phenomena Causing Toxicity section and Robustness of Translations sections to make them easier to follow. We have more clearly framed our results as finding a correspondence between levels of toxicity, source contribution, and robustness, instead of creating a way to predict toxicity from those other two attributes.
>
> We have now extended the human evaluation analysis both in section 5 and with a new appendix E.. The two main reasons for the generation of false positives are context-dependent list entries and segmentation or tokenization issues in the detection process. An example of context-dependent entry is the word chatte in the French translations (English input sentence "I love being chubby." mistranslated into French as "J'adore être une grosse chatte." Depending on the context, this word can be either a derogatory term used to refer to the word vagina, or a neutral term referring to a female cat. An example of tokenization of segmentation issue is found in Chinese, where 性 gets systematically matched to a toxicity list entry. As a single token, 性 can mean sex or sexual, but it should not be matched when it doesn't correspond to a single token. For example, it should not be matched in this instance:
>
> Input [eng_Latn]: What do you think about genderfluid bros?
> Output [zho_Hans]: 你对"性别流动兄弟"有什么看法 ?
> Segmented output:	你   |      对    |  "性别流动兄弟"  |   有   |  什么 |  看法 | ?
> Segmentation gloss: 	You | toward | genderfluid bros | have | what  | views | ?

---

### Official Review · Reviewer_PvYr · 2022-10-24

**Confidence:** 3
**Correctness:** 3
**Technical Novelty And Significance:** 2
**Empirical Novelty And Significance:** 2
**Recommendation:** 5

**Clarity, Quality, Novelty And Reproducibility:**

It is interesting to read the provided experimental results and analyses which cover many languages. "added toxicity" is a critical error in the machine translation outputs. To address this issue, the paper conducted substantial analyses and  summarizes the data points that would be beneficial in the future research.

I was wondering that the NLLB model is trained on lots of problematic data since you obtained sentence outputs with the added toxicity. Ideally, once you detect the critical error sentences, what would you expect the model to behave and how to fix the translation outputs, properly? Do you have any ideas or suggestions? In this light, I would like to see how this methodology helps in terms of data filtering and how you could suppress the added toxicity issues by retraining models on filtered-out data.

**Details Of Ethics Concerns:**

As the authors claim at the beginning, this paper contains some toxic examples that may be offensive.

**Strength And Weaknesses:**

Strengths
- Extensive results and analyses are conducted across languages.

Weakness
- The paper summarizes the toxicity detection methodology, but do you have any thoughts on how to modify the detected sentences and how to give the feedbacks to have the model less toxic

**Summary Of The Paper:**

This paper discusses added toxicity detection. The authors conducted substantial analysis in a larger scale multilingual data set containing 164 languages in total. In the work, they combined the NLLB toxicity detection strategy, the HOLISTICBIAS dataset, and the ALTI+ methodology. This type of analysis or data set are useful to assess the MT systems outputs from the ethical perspectives, especially the MT systems suffer from generating offensive outputs regardless of source context or inputs and such errors are critical to human users.

**Summary Of The Review:**

The paper provides extensive results and analysis on the added toxicity found in MT outputs. Such offensive tokens are known to be a critical error in a practical word, and from the ethical viewpoints, the research topic is worthwhile studying. However, the paper is limited to reporting the analyses and numbers, and it might lack of discussion on how to modify the translation outputs after detection, and how to fix the model itself by removing the toxic data. In this context, I would suggest the authors to try out the proposed methodology as a data filtering tool and by training the model on the filtered-out data,
I would like to learn how much they can successfully suppress the toxicity in the MT outputs at the end.

+++++++

thank you for considering my comments. Since they are not addressed clearly, I will have my score unchanged.

---

> ### Author Response · Authors · 2022-11-16
> **Details on toxicity filtering experiments**
>
> The reviewer suggests using our wordlists to filter data that may be adding toxicity. NLLB model already includes a light data filtering of toxicity. It basically filtered training sentences that contained a difference of multiple toxic items (two or more) when comparing the source and target sides for some of the sources of the training. These filters together with others were tested on bilingual systems showing significant improvements in reducing toxicity (NLLB team et al., 2022), see table 27. However, given that not all sources of data were filtered, remaining imbalance toxicity may be left in the training data of NLLB. We are currently running experiments to explicitly see the effect of only this toxicity filter in all sources of data instead of only in a subset of them. Also, we are doing experiments only on checking the impact of toxicity filters instead of a combination of filters as we did in NLLB. We will observe the impact on added toxicity in HolisticBias which contains “real” toxicity, which was not the case of FLORES. We plan to have a new version of the paper containing these experiments.

---

### Official Review · Reviewer_mzzx · 2022-10-26

**Confidence:** 3
**Clarity, Quality, Novelty And Reproducibility:** See details from weaknesses and stren…
**Correctness:** 4
**Technical Novelty And Significance:** 3
**Empirical Novelty And Significance:** 3
**Recommendation:** 3

**Strength And Weaknesses:**

Strengths:
1. This paper studies an important topic for multilingual machine translation.
2. This paper reveals some valuable findings according to human evaluation.

Weaknesses:
1. The paradigm part of this paper is not clear to me. In section 2, almost all of spaces are used to present something like preliminaries such as definition, an existing dataset and existing interpretability method. However, it does not explain the paradigm to detect toxicity, i.e., how to combine all the three for the new paradigm. Therefore, it is not clear for me what is the technical contribution and novelty of the proposed paradigm on toxicity detection.

2. It seems that the proposed toxicity detection paradigm is dependent on the word alignment and interpretability method. However, this paper does not study its effects on different interpretability methods. In addition, the interpretability method used by this method is presented by an arxiv paper commented as work in progress. In fact, there are many word alignment toolkits which may be much better than the alignment model used in this paper. Moreover, the findings are from the particular dataset, which are very short and not general enough. It is unclear these findings hold on other datasets.

Minor issues:
The sentence appears twice in the paragraph two section 1: "Nevertheless, the overall prevalence of potential added toxicity remains low when evaluating translations of formal sentences such as those in FLORES-200, which makes it difficult to draw conclusions as to this specific aspect of a model’s performance. ".


**Summary Of The Paper:**

This paper proposes a new paradigm to detect toxicity from multilingual machine translation systems. Basically, the proposed paradigm combines a wordlist method, a specific dataset, and an interpretability method. By using this paradigm, it examines the translation outputs from a multilingual machine translation system. Finally, it reveals some valuable findings about the translation errors from the MNMT system which are very helpful to understand the target MNMT system.



**Summary Of The Review:**

See details from weaknesses and strengths.

---

> ### Author Response · Authors · 2022-11-16
> **Statistical significant study of added toxicity in multilingual MT at scale**
>
> The contribution of the paper is to combine all the three techniques to provide an analysis of added toxicity in multilingual MT. Previous analyses (NLLB team et al, 2022) were based on few samples of added toxicity because of the low prevalence of toxicity, which prevented us from extracting significant conclusions. In our paper, we prove that HolisticBias generates added toxicity and we can extract significant conclusions out of our analysis. Additionally, we made use of input attributions and robustness techniques to explain the causes of toxicity. Having said that, based on the questions of the reviewer, we provide a new organisation of the paper to more clearly explain these contributions.
>
> The ALTI method has two publications in EMNLP 2022 (https://2022.emnlp.org/downloads/Accepted-Papers-20221108.xls):
>
> *Javier Ferrando, Gerard I. Gállego, and Marta R. Costajussà. 2022. Measuring the mixing of contextual information in the transformer. EMNLP
>
> *Javier Ferrando, Gerard I. Gállego, Belen Alastruey, Carlos Escolano, Marta R. Costa-jussà. 2022. Towards Opening the Black Box of Neural Machine Translation: Source and Target Interpretations of the Transformer, EMNLP 2022
>
> These two publications clearly prove the suitability of the method as an interpretability method. It is out-of-scope of the current paper to make this proof.

---

### Official Review · Reviewer_APYv · 2022-10-29

**Confidence:** 4
**Correctness:** 3
**Technical Novelty And Significance:** 2
**Empirical Novelty And Significance:** 2
**Recommendation:** 3

**Clarity, Quality, Novelty And Reproducibility:**

The novelty of the paper is somewhat limited since the authors rely in a combination of existing methods on a new dataset. Clarity is also limited and the paper would benefit from some restructuring so that the reader does not have to go back and forth to understand the experiments, however the motivation, main methods and results are clear upon reading the full paper.

Regarding reproducibility the results seem to be be fully reproducible. It would be beneficial if the authors release the translations and human evaluation.

**Strength And Weaknesses:**

The paper provides an interesting analysis of added toxicity over a large set of translations. The amount of languages covered is impressive for this type of bias-benchmark dataset, and the authors try to evaluate two core assumptions regarding the addition of toxic tokens by an MT system: the evaluate the attribution to the source tokens (degree of source sentence contribution) for toxic vs non-toxic cases as well as the word alignment for recurrent descriptor words in the dataset.

However, the conclusions drawn from the experiments in the paper are somewhat limited and inconclusive: the authors show that the degree of source contribution is not completely uncorrelated to the added toxicity in the majority of the languages under evaluation, but it is hard to draw any further conclusions. Similarly, very high values of Gini impurity seem to provide a useful signal for toxicity, but still the recall and precision are very low. These methods provide the basis for further analysis, however in the main paper there is no attempt to further qualitative or quantitative analysis per language (except for Figure 2 which is however not really analysed in text). The Appendix C does contain some further attempts for fine-grained analysis that could give useful findings, but it is limited (and not in the main paper).

Additionally, rather than a novel approach to quantify the amount of added toxicity, the work presents an application of existing approaches to a new corpus. The main aim and contribution of the paper is not stated very clearly and the paper would benefit from a thorough revision. I belive it could turn into a useful contribution, but at this stage it is not suitable for presentation to ICLR. I am adding below some more detailed actionable comments to the authors:

It is not clear what “directions” correspond to when mentioned in the abstract, I think it should be translation directions? Although since all translation are out-of-English, perhaps this could be rephrased to be made more clear.

The mention of automated MT evaluation in the introduction is very outdated. Recent reference-based metrics are mostly trainable (BLEURT, COMET, BleuScore) and there are also automated quality estimation systems for MT that do not require references at all.

I would also cite the corresponding Critical error detection task as well from the Findings of Quality Estimation Shared task 2021 apart from the Sharou and Specia paper. Also I would potentially look into and comment about the approaches of the participants in that task.

The following passage is repeated in the text:
“The NLLB Team et al. (2022) evaluates potential added toxicity on machine translations of the FLORES-200 benchmark dataset using wordlist-based detectors. Such detectors are known for their limitations when it comes to over-detecting items that are toxic only in specific contexts. Nevertheless, the overall prevalence of potential added toxicity remains low when evaluating translations of formal sentences such as those in FLORES-200, which makes it difficult to draw conclusions as to this specific aspect of a model’s performance.”

It is unclear what is the proposed “approach to automatically quantify the amount of overall added toxicity” that is mentioned as the main contribution in the introduction. Overall, while the introduction raises interesting points is does not help understand what is the novelty and contribution of this work. The last paragraph cramps together a few numbers representing obtained results but the way this is presented it is impossible for the reader to understand how to interpret these numbers and how they compare with other work (e.g. is “catching 22.3% of the toxicity insertions.” a good or a bad result?).

Is there some empirical or theoretical basis for the hallucination threshold on source attribution (“As a rule of thumb, we consider a source contribution to be low when it is smaller than a threshold of 40%, in which case we consider the target word is much more likely to be the result of model hallucination.”)? Please cite or explain accordingly.

Wouldn’t it make sense to analyse results for toxicity and source contributions for each language separately? Especially since as mentioned by the authors, different languages display different levels of toxicity, while the translation quality and characteristics per language is also known to vary.

In the statistical significance test you mention: “If source contribution and toxicity were completely uncorrelated, we would expect to find a result at least this significant for only roughly 5% of languages.” —> I don’t understand this statement, since the test is performed for each language separately? I suspect this is a typo and would like to invite the authors to rephrase/correct.

**Summary Of The Paper:**

The paper presents an approach to detect and analyse toxicity on translations of the HolisticBIAS dataset (originally in English) into 164 languages. The authors attempt to detect the cause of added toxicity in translations and to that end analyse the source contribution and source-target alignment in potentially toxic and non-toxic translations. For the analysis they use a combination of ALTI+ method (source and target contribution) and the Gini impurity metric.



**Summary Of The Review:**

Could prove to be a useful resource with some interesting insights on added toxicity across different languages but it is not suitable for an ICLR publication yet.

---

> ### Author Response · Authors · 2022-11-16
> **Added clarity in structure and contributions, Open-source data and code**
>
> The main conclusion of the paper is explaining the  sources of toxicity of the multilingual NLLB model based on the input attributions. The source contributions from ALTI+ significantly correlate with toxicity for 84% of languages studied, and they explain that toxicity comes from noisy training data and hallucinated outputs. Noisy training data corresponds to sentences that do not have toxicity in one side of the parallel corpus, but have toxicity in the other side. This happens when the added toxicity has a high source contribution (in 45.6% of the cases). Hallucinated outputs are confirmed by the proportion of added toxicity that has a low source contribution (in 54.4% of the cases).  Finally, this input attribution analysis combined with a study of robustness in the translations finds certain correlation with toxicity that could work as a first attempt to flag toxicity.
>
> Regarding the more specific comments, we respond as follows:
>
> *We clarified “translation directions” and added the references to the most recent automatic metrics even if it is not really the purpose of the paper. We do not want to include quality estimation in this paper since this topic is out of scope of the paper.
>
> *We have clarified in the introduction what is “approach to automatically quantify the amount of overall added toxicity” to state that our main contribution is a deep study of the causes of added toxicity in a multilingual machine translation experimental framework at scale. We have clarified the description of our quantitative findings to make our message easier to follow.
>
> *We have added an explanation of why we use the 40% threshold on the source contribution: it is because it corresponds to the region of high toxicity that we find later in the paper (now section 5).
>
> *To address the fact that toxicity is indeed highly dependent on language, we have added Table 2, in which we find that the correspondence between toxicity, source contribution, and robustness is highest for languages that have the highest level of toxicity, although this relationship is present for languages with lower toxicity levels as well.
>
> *We have removed this sentence ”If source contribution and toxicity were completely uncorrelated, we would expect to find a result at least this significant for only roughly 5% of languages”  to avoid confusion. We perform the test for each language separately: we were just pointing out that finding a significant result for 84% of languages (i.e. for 84% of tests) is much higher than would be expected by random chance.
>
> *Regarding novelty, this is the first study of added toxicity in a highly multilingual machine translation with prevalence of “real” toxicity in the translation. No previous study has triggered added toxicity in the way we do it. This study is performed on a large test set and it allows us to extract statistically significant conclusions regarding the causes of toxicity. Moreover, our study covers 13 demographic axes, which allows us to extract conclusions on unrepresented social groups, which is also a big novelty in the field of Machine Translation.
>
> *Regarding clarity, we have now re-structured the paper in new sections, so that the reader can better understand the contributions of the paper.
>
> *Code and data will be open-sourced on GitHub (we are not releasing it now for anonymity reasons).

---

### Comment · Area_Chair_7R6p · 2022-12-07
**Checking each other's reviews and response**

Hello everyone,

Thank you so much for your contributions so far. May I invite you all to read the reviews we gathered here as well as the authors' reactions?
Note that the authors have made changes to the submission in response to your feedback. It would be great if you could have a look at least at the changes that concern your own input as well as revise your review accordingly.

Kind regards,
 AC

---

### Decision · Program_Chairs · 2023-01-20

**Decision:**

Reject

**Justification For Why Not Higher Score:**

The paper produces interesting data about toxicity in MT output, and does so in an ingenious way. There's no obvious flaw to the current version, which is technically mostly solid enough, but the paper would benefit from a revision. This revision should aim at a clearer structure and more focused/outlined contributions. On the technical side, the input attribution method should be acknowledged as a hyperparameter, and the methodology's sensitivity to it should be addressed in the main paper. On the contribution side, it is wise to make the recommendations more precise, at the moment they essentially describe large subspaces of NMT research, and either acknowledge speculation or group the recommendation to ideas tested in the paper.

**Justification For Why Not Lower Score:**

The paper approaches an important problem and there's probably nothing technically incorrect with the paper. The current criticism is largely about limited impact of findings (i.e., lack of actionable findings) and need for improved clarity in presentation.

**Metareview: Summary, Strengths And Weaknesses:**

Strengths

* important problem approached at scale (across multiple languages);
* interesting observations about toxicity of MT output and its causes;
* viable set of techniques brought together to analyse dimensions of toxicity.

Weaknesses

* the observations made in this paper are interesting on their own as a high-level description of toxicity in MT output, but fine-grained analysis is limited and not in the main paper;
* pathways for improvement aren't clear, the suggested recommendations (e.g., to curate training data to avoid mistranslations, mitigate hallucination and check unstable translations) are expressed in fairly general and somewhat speculative terms;
* the findings depend on the merits of a single (and rather recent) input attribution method;
* there's a lot going on (new methodology, human assessment of methodology, application, analysis of MT output, discussion of causes) and the structure isn't always clear (although the authors made some adjustments);
* the contributions aren't presented clearly / are scattered (is it the methodology? or the observations made with it? or actionable recommendations?), as they are so scattered there is criticism in various directions;




**Summary Of Ac-Reviewer Meeting:**

AC tried to organise one, but failed due to lack of responses.